# Trend-based quality assurance of binary urinary antigen tests

Susanne Sütterlin[1,2*], Anders Olof Larsson[3]

**1** Department of Women's and Children's Health, Pediatric Nephrology and Infectious Diseases (PNID) Research Group, Uppsala University, Sweden, **2** Department of Pediatric Emergency, Children's Hospital, Uppsala University Hospital, Sweden, **3** Department of Medical Sciences, Section of Clinical Chemistry, Uppsala University, Sweden

\* susanne.sutterlin@uu.se

## Abstract

### Objective

Binary diagnostic assays, such as urinary antigen (uAg) tests, provide rapid results but lack quantitative output for conventional quality control. This study evaluated whether long-term monitoring of test positivity patterns can serve as a practical tool for quality assurance (QA) of pneumococcal and Legionella uAg assays.

### Methods

All pneumococcal and Legionella uAg tests performed in Uppsala County, Sweden (01/01/2007–31/12/2024), were retrospectively analyzed. Positivity trends were assessed over time, including seasonal variation and demographic subgroups. Outlier detection was performed using interquartile range (IQR) thresholds to identify potential analytical drift. Bayesian predictive values were additionally estimated as a theoretical comparison to illustrate the influence of prevalence on predictive performance.

### Results

A total of 17,356 pneumococcal and 15,280 Legionella uAg tests were included. Pneumococcal positivity displayed significant seasonal variation, lowest in August (OR 0.49; 95% CI 0.35–0.68), whereas Legionella positivity varied mainly by year, with peaks in 2007, 2012, and 2021–2022. Pneumococcal positivity was highest in children and female patients, while Legionella showed no demographic trends. QA intervals derived from IQR thresholds captured expected long-term stability (7.4% for pneumococcus; 0.7% for Legionella), with outliers corresponding to known epidemiological events. Bayesian estimates highlighted the large discrepancy between incidence-based and test-based predictive values but were secondary to the trend-based framework.

**Data availability statement:** Yes - all data are fully available without restriction; The minimal anonymized dataset required to reproduce the analyses is available in the Zenodo repository: https://doi.org/10.5281/zenodo.18989309.

**Funding:** The study was supported from the ALF agreement between Uppsala University and Region Uppsala (ALF-1037235). The funders had no role in study design, data collection and analysis, decision to publish, or preparation of the manuscript.

**Competing interests:** The authors declare that they have no competing interests.

**Abbreviations:** PPV, positive predictive value; NPV, negative predictive value; uAg, urinary antigen; CI, confidence interval; FOHM, Folkhälsomyndigheten: Swedish Public Health Agency.

### Conclusions

Rapid uAg tests pose challenges for QA. Positivity trends provide a feasible strategy for long-term, population-based monitoring of binary diagnostic tests and can complement conventional controls and detect analytical drift.

---

## Background

Urine antigen (uAg) assays for Streptococcus pneumoniae and Legionella pneumophila are widely used in the rapid diagnostic work-up of community-acquired pneumonia [1,2]. These assays yield binary results—positive or negative—without information about how close a result is to the decision threshold. Therefore, uAg test results may be subject to subtle shifts in sensitivity or specificity that can go undetected over time [3,4].

An alternative strategy is to use patient-derived results for method monitoring. Patient medians are used in clinical chemistry, for example in HbA1c, to detect long-term shifts in analytical performance that are not captured by synthetic control materials [5]. In the same way, monitoring positivity trends in binary assays can provide valuable information on assay stability and potential drift. Robust statistical tools such as the interquartile range (IQR) can be applied to detect outliers and define expected intervals for routine performance, even in non-normally distributed real-world data.

In contrast to routine clinical chemistry parameters, pneumococcal and Legionella infections are strongly influenced by infectious disease epidemiology, creating the need for context-aware QA strategies. Both pathogens are relatively uncommon at the population level, with incidences of approximately 15 per 100,000 for invasive pneumococcal disease and 1–2 per 100,000 for Legionella in Sweden [6,7]. Low disease prevalence inevitably lowers the positive predictive value (PPV) of diagnostic assays, a well-known consequence of Bayes' theorem. While Bayesian estimates can therefore provide useful theoretical context, their practical application in QA is limited by the absence of true prevalence data for the tested population.

Taken together, these considerations suggest that long-term monitoring of positivity trends offers a feasible, patient-based complement to conventional QA for binary assays such as urinary antigen tests. In this study, we retrospectively analyzed pneumococcal and Legionella uAg results over 17 years, applying trend analysis, seasonal comparisons, and IQR-based thresholds to evaluate test stability over time. Bayesian estimates were included as a theoretical comparison to illustrate the influence of prevalence, but the main focus was on trend-based approaches to quality assurance.

## Materials and methods

### Samples and setting

All uAg tests for pneumococcal and Legionella uAg requested in Uppsala County during the study period were analyzed at the Department of Clinical Chemistry at Uppsala University Hospital and included all healthcare providers across Uppsala

County, including primary care centers, Enköping County Hospital, and Uppsala University Hospital. Samples were collected in sterile, additive-free containers and were promptly transported to the laboratory. All test results were recorded in the laboratory information system.

Anonymized data were extracted for all tests performed between 01/01/2007 and 31/12/2024, including test result, sample date, patient age, and sex. Information on age and sex was complete for all samples, however no clinical outcome data or treatment information was accessible. Data were accessed for research purposes on 13/04/2025. Analyses were sample-based: repeated tests from the same individual/episode were not removed, as the objective was assay-level quality monitoring. The study was approved by the Regional Ethical Review Board in Uppsala (Dnr 01–367), informed consent was not required.

### Analysis

Pneumococcal and Legionella antigens were analyzed using the BinaxNOW Streptococcus pneumoniae Antigen Test and the BinaxNOW Legionella uAg Test (both Abbott, USA), following the manufacturer's instructions. Both assays are based on immunochromatographic lateral flow technology that detects specific bacterial antigens in urine samples. All tests were performed at room temperature using fresh urine, and test results were read within 15 minutes using the Alere DIGIVAL Reader (Abbott, USA) between 01/01/2007 and 27/06/2024, and the Abbott DIGIVAL from 28/06/2024 onwards. Results were recorded as binary outcomes: positive or negative.

### Statistical analyses

All statistical analyses and visualizations were performed in R (version 4.5.0, R Foundation for Statistical Computing), and plots were created using the *ggplot2* package. The dataset used was sample-based in order to reflect routine laboratory workflow.

Descriptive statistics were used to summarize the number of uAg tests performed and the proportion of positive results over time. Age categories were pre-specified on clinical and epidemiological grounds – pediatric (0–19), working-age adults (20–59), older adults (60–79), and the very old (≥80 years), and analyses were stratified by age group and sex. Proportions with 95% confidence intervals were calculated using binomial tests (prop.test in R). Seasonal variation in positivity was analyzed with logistic regression with calendar month entered as a 12-level categorical factor (January as reference), reporting odds ratios with 95% confidence intervals. Trends in annual test volumes were estimated using linear regression, with slope, 95% confidence intervals, and p-values reported. Correlation between monthly test volume and positivity was assessed with Pearson's correlation on monthly aggregates. As a theoretical comparison, annual PPV and NPV were estimated using Bayes' theorem with reported sensitivity and specificity values. Calculations were performed both with national incidence data and with observed test positivity as a proxy for prevalence. These estimates were included to illustrate the effect of prevalence on predictive values.

For the BinaxNOW *S. pneumoniae* uAg test, sensitivity and specificity were set to 86% and 94%, respectively, according with the manufacturer. For the BinaxNOW Legionella test, both sensitivity and specificity were set at 95% in accordance to the manufacturer. National incidence data were used as a conservative proxy prior for disease prevalence in Bayes' formula; no transformation by assumed episode duration was applied. In contrast, observed test positivity was treated as a point prevalence of antigen detection among tested patients. PPV and NPV were computed twice: first using national incidence for invasive pneumococcal disease and Legionella disease as reported by the Swedish Public Health Agency [6,8], as both conditions are notifiable in Sweden. Then using observed test positivity as the prior to better reflect the tested population. The observed positivity (test-based prior) was used as an operational quality assurance marker and may be subject to test-based prevalence bias, as the same assay informs both prior and likelihood. As a population-level comparator, predictive values were also computed using national incidence as a conservative lower-bound prior.

Assay performance drift – meaning unintended shifts in effective sensitivity or specificity due to lot-to-lot variation, reagent aging, storage, or visual interpretation – was evaluated using quality-assurance control thresholds. The manufacturer's qualitative decision thresholds for positive/negative results were neither altered nor re-estimated, instead, population outputs were monitored. Quality assurance thresholds for positivity rates were defined using the interquartile range (IQR) method, flagging annual or monthly values outside the median ±1.5×IQR as potential signals of analytical drift [9].

## Results

### Test volume over time

A total of 17,355 pneumococcal uAg and 15,279 Legionella uAg test results were analyzed. Female patients accounted for 45.6% (7,913/17,355) of pneumococcal tests and 43.5% (6,648/15,279) of Legionella tests. The median age for both groups was 72 years (Q1-Q3: 56–81 years). Despite interannual variation, the number of tests increased significantly over time for both antigens. For pneumococcal uAg, the annual increase was calculated as 69 tests (95% CI: 53–85; $p < 0.001$; linear regression), corresponding to an approximate 7% relative increase per year. For Legionella uAg, the increase was 74 tests per year (95% CI: 59–89; $p < 0.001$; linear regression). (Fig 1, top panel)

### Annual positivity rates and seasonal variation

The median annual positivity rate was 7.4% (IQR 6.4%–9.0%) for pneumococcal uAg and 0.7% (IQR 0.5%–0.8%) for Legionella uAg. While the pneumococcal positivity rate showed limited interannual variation—except for the first study year (2007)—the Legionella positivity rate exceeded the 1.5×IQR threshold in 2012 and again in 2021–2022. Although overall test volumes increased over time, the distribution of tests across months remained relatively even within each year (Fig 1).

Seasonal variation in positivity was assessed using logistic regression with January as the reference month. For pneumococcal uAg, a distinct seasonal pattern was observed, with significantly lower odds of a positive result from June to November. The lowest odds ratio was seen in August (OR 0.49, 95% CI 0.35–0.68, $p < 0.001$). In contrast, no statistically significant seasonal trend was observed for Legionella uAg; however, August yielded the highest odds ratio (OR 2.07, 95% CI 0.96–4.72, $p = 0.069$), suggesting a possible summer peak in some years (Fig 2).

The number of pneumococcal uAg tests also varied seasonally, with lower volumes during summer months. The median monthly test volume ranged from 58 tests in June (IQR 50–92) to 92 tests in March (IQR 61–106). For Legionella uAg, similar seasonal fluctuation was observed. Over the study period, the median monthly number of tests ranged from 49 (IQR 40–83) in June to 72 (IQR 49–91) in January, indicating year-round variability in test frequency (Fig 2).

No significant correlation was found between monthly test volume and positivity rates for either pneumococcal uAg ($r = -0.02$, $p = 0.78$) or Legionella uAg ($r = 0.11$, $p = 0.097$), indicating that positivity remained stable regardless of testing frequency. This supports the interpretation that test positivity can be evaluated independently of changes in testing intensity, strengthening its potential use as a quality assurance indicator. (Fig 3)

### Monthly variation of test positivity stratified by age and sex

For pneumococcal uAg testing, the positivity rate declined with increasing age, from 20.3% in the 0–19 age group (IQR 17.7–23.1%) to 5.3% in individuals aged ≥80 years (IQR 4.7–5.9%). In contrast, positivity rates for Legionella uAg remained consistently low across all age groups, ranging from 0.5% in children and adolescents (0–19 years) to 1.0% in adults aged 60–79 years, with overlapping interquartile ranges and no apparent age-related trend (Table 1).

Among those tested for pneumococcal uAg, females exhibited a significantly higher positivity rate compared to males (8.6% vs. 6.3%; $p < 0.001$), with interquartile ranges of 8.0–9.2% and 5.9–6.9%, respectively. For Legionella uAg, positivity was similarly low in both sexes: 0.8% (IQR 0.6–1.1%) in females and 0.9% (IQR 0.7–1.1%, $p = 0.7$) in males, with no statistically significant difference (Table 1).

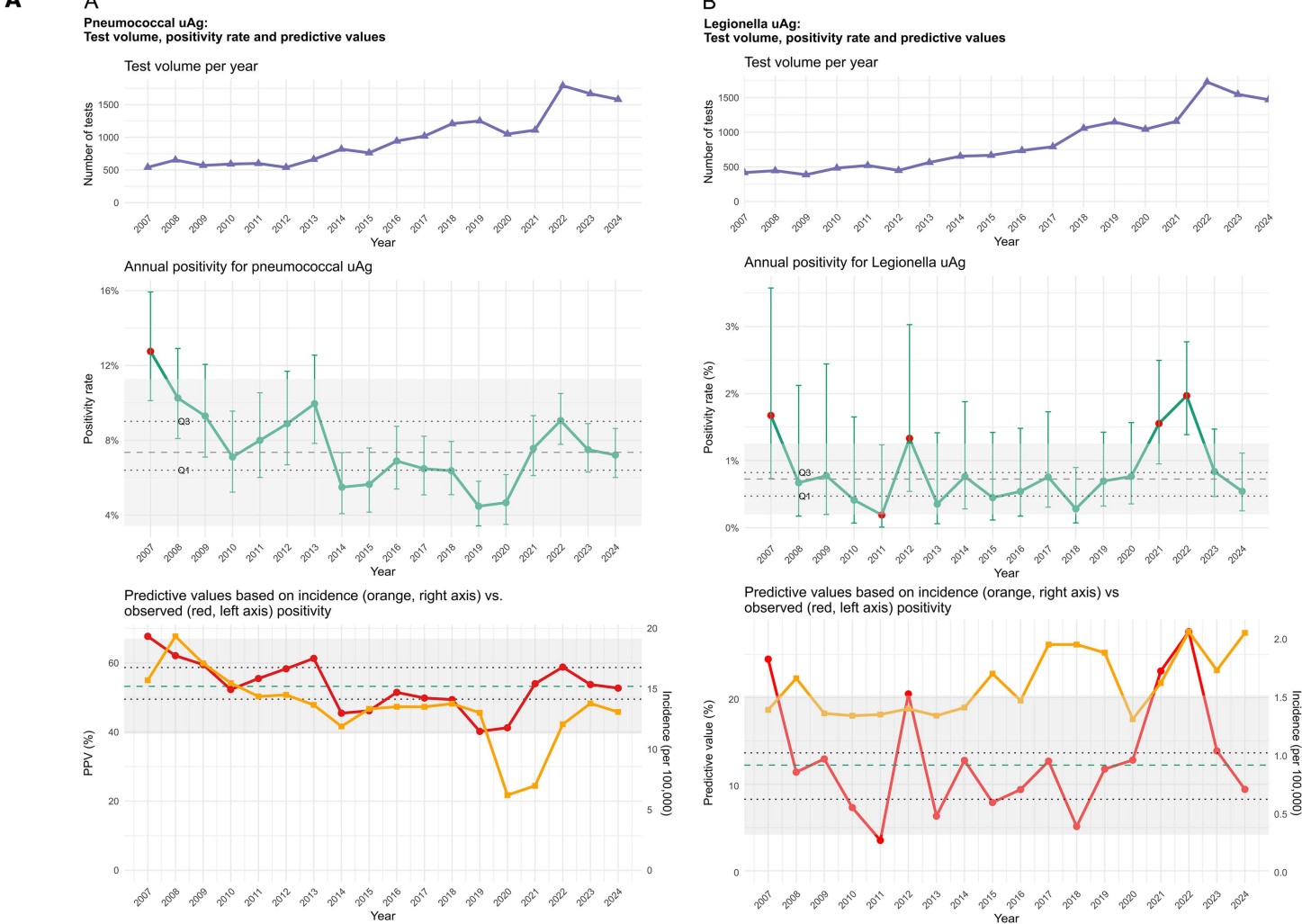

**Fig 1. Annual test volume, positivity for and contextual predictive value estimates for pneumococcal (panel A) and Legionella (panel B) urinary antigen (uAg) tests, 2007-2024.** The upper panel shows the annual number of tests performed. The middle panel presents the observed annual positivity rate with 95% binomial confidence intervals, median positivity (dashed line), interquartile range (Q1–Q3, dotted lines), and the shaded area indicating the Tukey-defined limits (median ±1.5×IQR). The lower panels display contextual Bayesian estimates of the positive predictive value (PPV: red, left axis) and the incidence of the reported invasive pneumococcal disease (1a) and Legionella disease (1b; incidence orange, right axis) according to the Swedish public health authorities, respectively. The shaded area indicates the IQR-based variation around the observed PPV, suggesting a potential data-driven interval for internal quality assurance measurement. Outliers correspond to known epidemiological events.

## Trend-based QA and contextual estimates

Quality assurance intervals for lab-based PPV were defined using the interquartile range (IQR) method. For pneumococcal uAg, the quality assurance range was calculated as 39.5% to 67.0%, and for Legionella uAg, 4.2% to 20.2%. The findings underscore the difference between test-based and incidence-based predictive values, highlighting the need for context-aware approaches in the quality monitoring of binary diagnostic assays.

For pneumococcal uAg testing, the median PPV based on reported incidence of invasive pneumococcal disease was 0.19% (range 0.08–0.28%) during the study period. When observed test positivity was used as a proxy for disease incidence, the median PPV increased markedly to 53.2% (range 49.5–59.7%). For Legionella uAg, the median PPV based on

**A** Monthly positivity for pneumococcal uAg (per year)

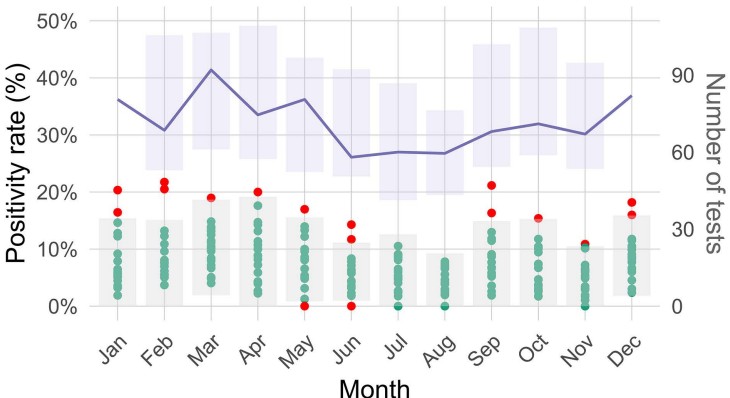

**B** Monthly positivity for Legionella uAg (per year)

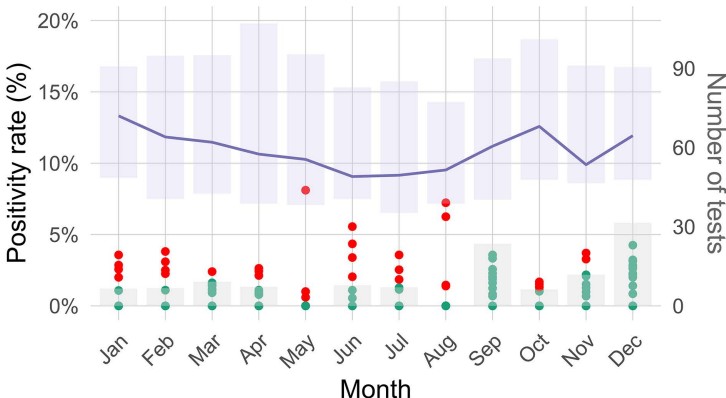

**Fig 2. Monthly positivity for pneumococcal (panel A) and Legionella (panel B) urine antigen tests, stratified by calendar month across study years.** Each dot represents the proportion of positive tests for a given month and year. Red dots indicate outliers, defined as values falling outside the expected range (gray band), calculated as the monthly median ± 1.5 × IQR. The purple band and line represent the interquartile range and median of the monthly test volume, respectively, with its y-axis on the right.

national incidence data was 0.03% (range 0.025–0.036%), while the lab-based Bayesian estimate yielded a substantially higher median PPV of 12.2% (range 8.3–13.6%).

Negative predictive values were high across all models. For pneumococcal uAg, the test-based median NPV was 98.85% (range 98.52–99.0%), while for Legionella uAg, the NPV remained at 100% throughout. Incidence-based NPV estimates were also 100% for both assays across all years.

## Discussion

This study evaluated long-term quality monitoring strategies for binary urinary antigen (uAg) tests used for Streptococcus pneumoniae and Legionella pneumophila. By analyzing more than 30,000 test results over 17 years, we show that positivity trends provide a stable and informative complement to conventional quality assurance (QA) systems. Pneumococcal positivity displayed clear seasonal variation, while Legionella positivity fluctuated mainly by year and corresponded to recognized epidemiological peaks. These findings support the use of patient-derived data as a practical tool for detecting potential analytical drift in binary diagnostic assays.

A    Correlation between number of tests and positivity rate for pneumococcal uAg

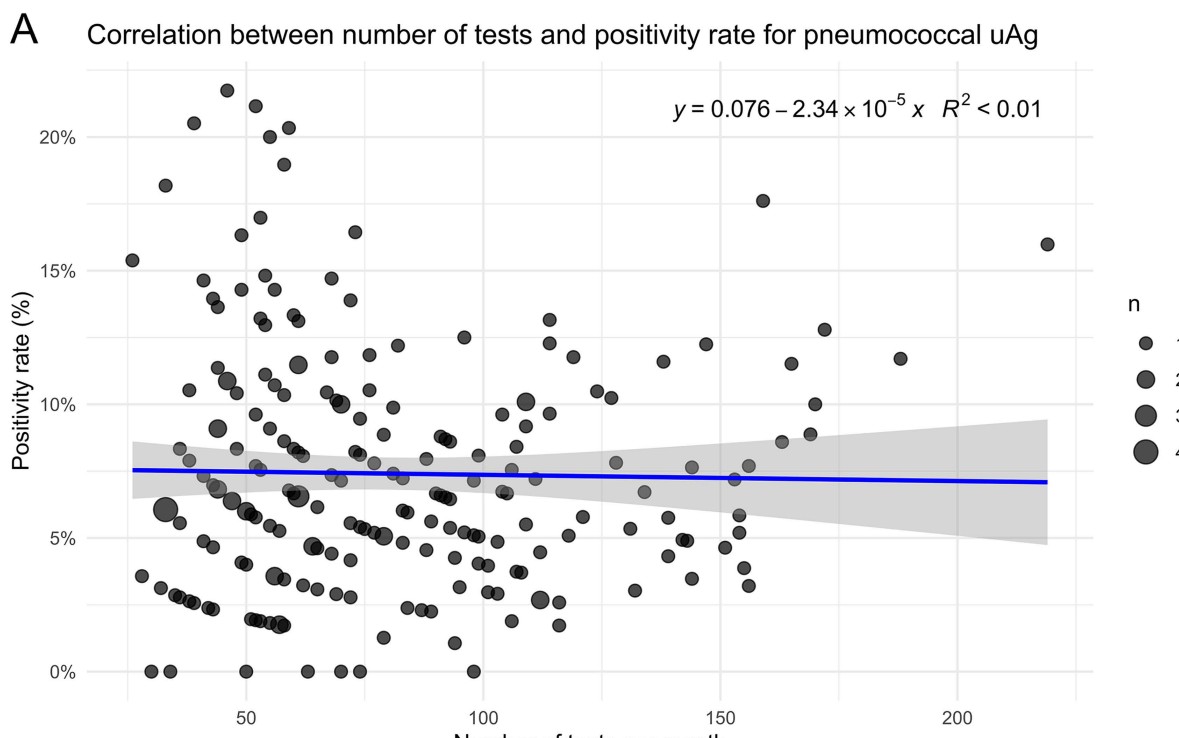

$$y = 0.076 - 2.34 \times 10^{-5}\, x \quad R^2 < 0.01$$

B    Correlation between number of tests and positivity rate for Legionella uAg

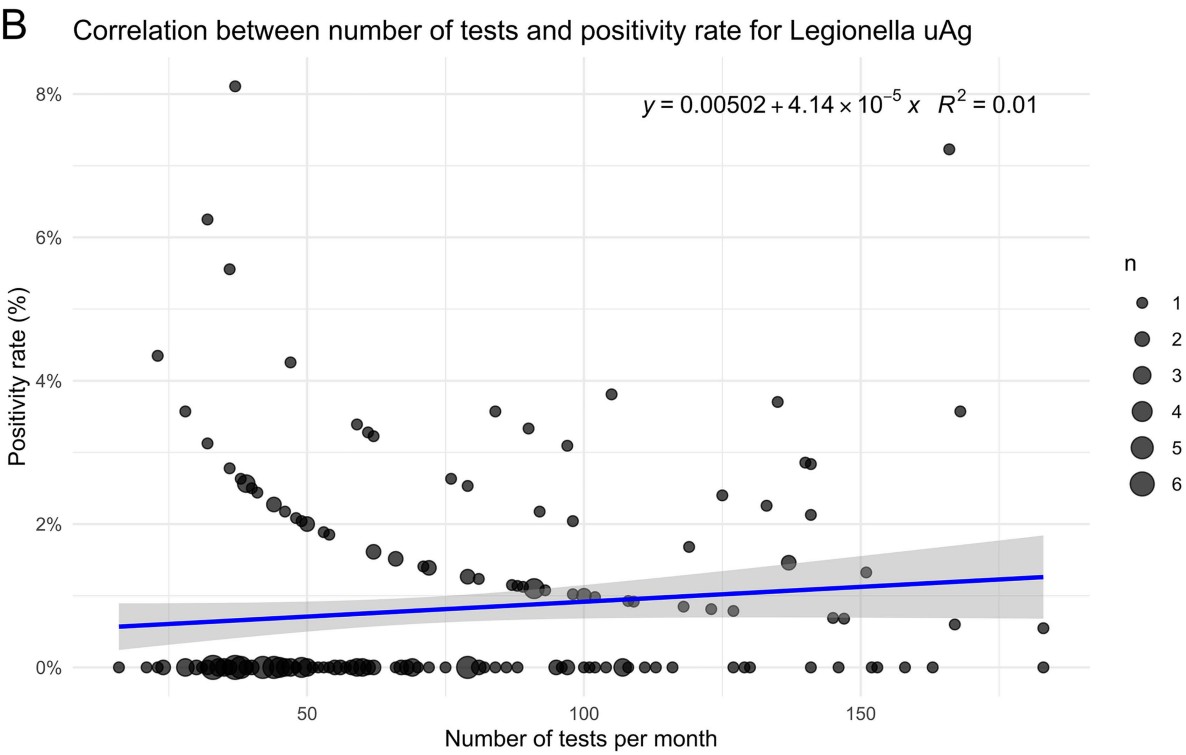

$$y = 0.00502 + 4.14 \times 10^{-5}\, x \quad R^2 = 0.01$$

**Fig 3. Scatterplots showing the correlation between the number of urine antigen tests performed per month and the proportion of positive results for pneumococcal (panel A) and Legionella (panel B) urine antigens.** Each dot represents one calendar month, dot size indicates months with the same combination of test volume and positivity rate. A linear regression line with 95% confidence band is shown.

**Table 1. Positivity rates for pneumococcal and Legionella urine antigen tests per age and sex.**

| | Pneumococcal urine antigen (Tested/total (%)) | Pneumococcal urine antigen (positive test/total (%)) | Lower – higher quartile | Legionella urine antigen (tested/total (%)) | Legionella urine antigen (positive test/total (%)) | Lower – higher quartile |
|---|---|---|---|---|---|---|
| **Age group (years)** | | | | | | |
| 0-19 | 878/17,355 (5.1) | 178/878 (20.3) | 17.7-23.1 | 410/15,279 (2.6) | 2/410 (0.5) | 0.1-1.9 |
| 20-59 | 4,061/17,355 (23.3) | 307/4,061 (7.6) | 6.8-8.4 | 3,715/15,279 (24.3) | 29/3,715 (0.8) | 0.5-1.1 |
| 60-79 | 7,255/17,355 (41.8) | 521/7,255 (7.2) | 6.6-7.8 | 6,915/15,279 (45,3) | 67/6,915 (1.0) | 0.8-1.2 |
| 80+ | 5,161/17,355 (29.7) | 273/5,161 (5.3) | 4.7-5.9 | 4,239/15,279 (27.7) | 36/4,239 (0.8) | 0.6-1.2 |
| **Sex** | | | | | | |
| Female | 7,913/17,355 (45.6) | 680/7,913 (8.6) | 8.0-9.2 | 6,648/15,279 (43.5) | 56/6,648 (0.8) | 0.6-1.1 |
| Male | 9,442/17,355 (54.4) | 599/9,442 (6.3) | 5.9-6.9 | 8,631/15,279 (56.5) | 78/8,631 (0.9) | 0.7-1.1 |

Binary diagnostic assays such as uAg tests rely on colorimetric, visually interpreted results and produce immediate binary outcomes without measurable gradation in signal intensity. Unlike assays with quantitative outputs that allow for the use of statistical control tools like Shewhart charts, uAg tests offer no continuous values that can be monitored for drift or instability [10]. Despite automation, the lack of a continuous output means that the range near the decision threshold where false positives and false negatives are most likely to occur—remains unquantified and thus undetectable through conventional quality control strategies. As described by Ríos *et al* this region is especially difficult to define in qualitative tests with binary outputs, but its implications for diagnostic accuracy and quality assurance are nonetheless significant [11]. While the use of internal positive and negative controls helps detect technical failures, these do not detect gradual shifts in test performance. In the absence of calibration curves or threshold-based metrics, population-based strategies, such as trend monitoring of positivity rates, become a essential tools for quality assurance over time [12]. An additional limitation is that internal positive and negative controls are supplied by the same manufacturer as the test kits. This may introduce a risk of manufacturer bias, as internal controls are inherently aligned with the test system and may fail to detect subtle analytical drift.

Another key challenge in monitoring the analytical performance of pneumococcal and Legionella uAg tests is the low incidence of these infections, which results in markedly reduced positive predictive values (PPVs)—a well-known consequence of Bayes' theorem in low-prevalence conditions [4]. As noted by Sinclair et al. (2013), the pooled sensitivity of the BinaxNOW S. pneumoniae test across 27 studies was 74.0%, with a pooled specificity of 97.2% for community-acquired pneumonia. These estimates contrast with the higher sensitivity reported by the manufacturer, based on studies of invasive pneumococcal disease. Furthermore, BinaxNOW S. pneumoniae is known to cross-react with other alpha-hemolytic streptococci, and BinaxNOW Legionella only detects L. pneumophila serogroup 1. These factors limit specificity, as demonstrated in multiple studies [13,14]; and introduce background noise and potential bias into longitudinal positivity trends, particularly if the circulation of non-target organisms or non-detected serogroups varies over time. In assays with binary readouts, such background variability increases vulnerability to small shifts in algorithmic decision thresholds, which could mimic epidemiological changes if not interpreted in context. Accordingly, observed positivity rates should not be interpreted as direct measures of pathogen incidence, but as composite signals shaped by assay biology, analytical thresholds, and epidemiological context. This discrepancy underscores the importance of context-specific performance monitoring, which is, however, limited by the absence of continuous surveillance for community-acquired pneumonia [15].

Trend-based quality monitoring, as presented in this study, differs from established external proficiency testing frameworks—such as the a-score system proposed by *Beavis et al.*—by not relying on replicate samples or inter-laboratory consensus [16]. Instead, we use longitudinal trends in positivity rates as internal indicators of analytical

consistency over time. In the here presented material, pneumococcal uAg positivity remained stable around 7% and L. pneumophila around 0.7% over 17 years, despite increasing test volumes and changing patient demographics. Detection methods such as IQR-based criteria have been described to suit non-normally distributed real-world data, supporting our use of interquartile positivity thresholds as quality indicators [9]. Notably, short-term increases in positivity (e.g., Legionella in 2021–2022) likely reflected national case clustering, indicating an epidemiological shift rather than problems in the analytical process. However, as such shifts can influence trends in positivity and predictive values, any outliers must be thoroughly investigated, including both test performance and the pathogen's actual epidemiology. [17].

Bayesian predictive values were included only as a contextual illustration, highlighting the expected difference between incidence-based and test-based estimates. Importantly, the PPV analyses were not intended to validate diagnostic accuracy, but to illustrate the strong dependence of PPV on prevalence assumptions over time. Population-level incidence data were therefore included as an external epidemiological reference, allowing assessment of whether temporal changes in PPV estimates aligned with known epidemiological shifts or outbreak-related events, rather than as a directly comparable prevalence measure for the tested population. Within this framework, deviations from IQR-defined ranges—for example, the increase in Legionella positivity in 2021–2022—most likely reflected true epidemiological shifts rather than analytical drift. Such outliers underline the importance of interpreting QA signals in light of both assay performance and pathogen epidemiology. Observed test positivity serves as a longitudinal monitoring signal rather than a surrogate gold standard, and the resulting PPV estimates should be interpreted as illustrative and context-dependent.

Pneumococcal infections display marked seasonal variation, whereas Legionella shows more year-to-year fluctuations, often linked to outbreaks. The peak in pneumococcal positivity observed in 2007 likely reflects pre-vaccine epidemiology, as national pneumococcal vaccination was introduced in 2009, with some regions implementing it already in 2008. This early peak falls outside the 1.5×IQR range and may be attributable to the absence of widespread vaccination at that time. Within the proposed framework, such structural breaks—most notably vaccine introduction—should be interpreted as epidemiological shifts rather than analytical drift, with baseline expectations assessed dynamically over time. In contrast, Legionella peaks in 2021–2022 coincide with post-pandemic period and align with national surveillance data from the Swedish Public Health Agency (FOHM). Similarly, increases in 2012 and in 2007 reflect a higher incidence of reported and travel-associated cases, respectively, supporting the interpretation that peaks in test positivity often mirror real epidemiological shifts rather than analytical variability.

While the literature reports a general male predominance among hospitalized patients with pneumococcal pneumonia, the here presented data showed a higher proportion of females with positive uAg tests. Crabtree et al. (1999) noted higher pneumonia-associated mortality in women despite comparable illness severity and time to treatment. In our material, women accounted for only 45% of all pneumococcal test submissions, suggesting that they may be tested later in the disease course or present with more advanced symptoms—resulting in a higher proportion of positive test results compared to men [18–20]. However, as the present analysis is based on aggregated laboratory data without linkage to individual clinical presentation or disease severity, such demographic findings cannot be disentangled from differences in test ordering behavior.

The most important limitation of this study is the absence of linkage between aggregated laboratory positivity and individual clinical outcomes, treatment information, or confirmatory reference testing. As the analysis is observational and based on aggregated laboratory data, validation against clinical disease burden or individual-level diagnostic accuracy is not possible, and interpretations are therefore restricted to population-level trends. The long observation period of 17 years and the large volume of uAg test data nevertheless allow for robust analysis of long-term trends and detection of deviations. The use of laboratory-based data reflects realistic diagnostic practice and minimizes bias related to clinical decision-making. Observed variation in test positivity was interpreted in light of known national epidemiological patterns, adding contextual validity to the findings. Bayes priors were approximated using national incidence data, included only as a conservative lower bound and subject to under-ascertainment in mandatory notification systems. Observed test positivity used was used as a monitoring signal rather than a surrogate gold standard, acknowledging the potential

 

for incorporation bias when the same assay informs both prior and likelihood. Accordingly, the proposed framework is intended for quality monitoring and early signal detection rather than definitive epidemiological attribution or diagnostic accuracy assessment. Nonetheless, the proposed framework offers a feasible and informative approach to long-term analytical quality assurance for binary diagnostic tests. While it is constrained by binary test outputs, access to quantitative diagnostic signals could enable earlier detection of analytical drift and improve discrimination between threshold effects and true epidemiological change, strengthening long-term quality assurance.

## Conclusions

This study demonstrates that long-term monitoring of test positivity provides a feasible and patient-based strategy for quality assurance of binary uAg tests. Using interquartile range (IQR)-defined thresholds, assay stability could be confirmed over 17 years, while outliers reflected known epidemiological events. This trend-based approach complements conventional internal and external controls and supports the detection of analytical drift in routine practice.

## Author contributions

**Conceptualization:** Susanne Sütterlin, Anders Olof Larsson.

**Data curation:** Susanne Sütterlin, Anders Olof Larsson.

**Formal analysis:** Susanne Sütterlin, Anders Olof Larsson.

**Methodology:** Susanne Sütterlin, Anders Olof Larsson.

**Project administration:** Susanne Sütterlin.

**Resources:** Susanne Sütterlin, Anders Olof Larsson.

**Software:** Susanne Sütterlin, Anders Olof Larsson.

**Validation:** Susanne Sütterlin.

**Visualization:** Susanne Sütterlin, Anders Olof Larsson.

**Writing – original draft:** Susanne Sütterlin, Anders Olof Larsson.

**Writing – review & editing:** Susanne Sütterlin, Anders Olof Larsson.

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
