## [Decision Letter · Decision Letter 0]

5 Jan 2026

PLOS One

Dear Dr. Sütterlin,

Thank you for submitting your manuscript to PLOS ONE. After careful consideration, we feel that it has merit but does not fully meet PLOS One’s publication criteria as it currently stands. Therefore, we invite you to submit a revised version of the manuscript that addresses the points raised during the review process.

I would like to sincerely apologise for the delay you have incurred with your submission. It has been exceptionally difficult to secure reviewers to evaluate your study. We have now received two completed reviews; the comments are available below. Reviewer #2 raised significant scientific concerns about the study that need to be addressed in a revision.

Please revise the manuscript to address all the reviewer's comments in a point-by-point response in order to ensure it is meeting the journal's publication criteria. Please note that the revised manuscript will need to undergo further review, we thus cannot at this point anticipate the outcome of the evaluation process.

We look forward to receiving your revised manuscript.

Kind regards,

Miquel Vall-llosera Camps

Senior Staff Editor

PLOS One

Journal Requirements:

4. In the online submission form, you indicated that the datasets used and/or analysed during the current study are available from the corresponding author on reasonable request.

6. Thank you for stating the following financial disclosure:

The study was supported from the ALF agreement between Uppsala University and Region Uppsala.

Reviewers' comments:

Reviewer's Responses to Questions

**Comments to the Author**

1. Is the manuscript technically sound, and do the data support the conclusions?

Reviewer #1: Yes

Reviewer #2: Yes

2. Has the statistical analysis been performed appropriately and rigorously?

Reviewer #1: Yes

Reviewer #2: Yes

3. Have the authors made all data underlying the findings in their manuscript fully available?

Reviewer #1: Yes

Reviewer #2: Yes

4. Is the manuscript presented in an intelligible fashion and written in standard English?

Reviewer #1: Yes

Reviewer #2: Yes

Reviewer #1: Binary tests (positive/negative) lack the quantitative output typically used in statistical process control (like Shewhart charts). This study addresses a significant gap by using population-level positivity trends as a proxy for assay stability.

Substantial Dataset: The use of over 32,000 tests (17,356 pneumococcal and 15,280 Legionella) across nearly two decades provides high statistical power to distinguish between technical drift and actual epidemiological shifts.

Context-Aware Interpretation: The researchers successfully correlated outliers (e.g., Legionella peaks in 2021–2022) with known epidemiological events rather than dismiss them as test failures, demonstrating the practical utility of their framework in a real-world setting.

Demographic Insights: The study uncovered significant demographic trends, such as higher pneumococcal positivity in children (20.3%) and female patients (8.6% vs. 6.3% in males), which adds secondary value to the primary QA objective.

Methodological Considerations

Statistical Robustness: The application of interquartile range (IQR) thresholds (median ±1.5×IQR) is an appropriate choice for "real-world" data that may not follow a normal distribution.

Theoretical vs. Practical Comparison: By including Bayesian predictive values, the authors illustrate the dramatic impact of prevalence on test performance, providing a useful theoretical benchmark for why lab-based monitoring is necessary.

Limitation - Lack of Clinical Linkage: A noted limitation is the absence of individual clinical outcomes. As the study is based on aggregated laboratory data, it cannot directly validate individual test accuracy against a "gold standard" clinical diagnosis.

Conclusion & Impact

The research is highly relevant for clinical laboratories seeking low-cost, effective ways to monitor qualitative assays. It provides a feasible strategy for long-term population-based monitoring that can detect subtle analytical shifts which internal manufacturer controls might miss. The study is well-structured and scientifically sound, offering a practical framework that can be adapted for other binary diagnostic assays beyond urinary antigens.

Reviewer #2: This manuscript addresses an important and underdeveloped aspect of diagnostic medicine, namely how to ensure long-term quality assurance for rapid, reader-based assays that provide only binary outcomes. By analyzing 17 years of laboratory testing data (2007–2024) comprising more than 30,000 assays, the study proposes a practical framework in which population-level longitudinal positivity trends are monitored to identify potential analytical drift and to distinguish true epidemiological variation from technical or operational failures. The work is potentially valuable for Sweden’s healthcare system because it offers stable long-term reference baselines for urinary antigen testing, with reported positivity levels of approximately 7.4% for Streptococcus pneumoniae and 0.7% for Legionella, which could serve as performance anchors for surveillance and rapid troubleshooting.

A central contribution is the demonstration that long-term positivity rates can remain stable despite large increases in test volumes over time. This stability supports the concept that routine clinical data can be used as an independent performance signal, particularly in settings where conventional quality controls may not detect subtle changes in assay behavior. The manuscript also carries a preventive medicine implication by suggesting that national policy interventions, specifically the 2009 introduction of pneumococcal vaccination, coincided with a measurable stabilization of pneumococcal test positivity compared with earlier years. The temporal patterns further underscore the need for seasonal and demographic clinical awareness: pneumococcal positivity appears lowest in August, whereas Legionella positivity varies unpredictably across years, supporting a rationale for continuous longitudinal surveillance to protect vulnerable groups.

The manuscript is especially relevant because it confronts the “black box” problem inherent to binary readouts. Because these assays do not provide graded signal intensity or proximity to the positivity threshold, standard statistical process control approaches are difficult to apply. The authors’ proposal to monitor population-level positivity trends is therefore an innovative and feasible strategy to detect drift or discontinuities that might otherwise go unnoticed. Related to this, the framework may mitigate potential manufacturer bias that can arise when internal controls are supplied by the same manufacturer that produces the test system. By leveraging patient-derived trends and summary statistics, the approach can function as a real-world, independent performance check that complements rather than replaces conventional controls.

Several points would benefit from clarification or strengthening in a revised version. First, the figures appear to be low resolution and are difficult to read; higher-resolution or vector-quality figures are needed so that trends, outliers, and subgroup comparisons can be evaluated reliably. Second, the manuscript should more clearly justify the methodological choice of using “observed test positivity” as a prior input when estimating positive predictive values. Because PPV is extremely sensitive to prevalence assumptions, the rationale for treating observed positivity as a proxy for prevalence should be explained in detail, including how the approach avoids circularity and what interpretation the resulting PPV is intended to support. The large discrepancy reported between incidence-based PPV estimates and test-based PPV estimates is striking and deserves careful discussion. The authors should clarify whether the incidence-based approach is using population-level incidence that is not directly comparable to the clinically enriched tested population, and they should contextualize any manufacturer-reported PPV values by describing the underlying study population, reference standard, and case definition. If manufacturer PPV appears low under certain assumptions, it would be helpful to explain whether this reflects test performance limitations or instead reflects the use of prevalence inputs that do not match the intended-use population.

The demographic findings also merit deeper consideration. The observation of higher pneumococcal positivity in females than males contrasts with some published reports suggesting male predominance among hospitalized pneumonia cohorts. While the authors propose that differences in the timing of testing might contribute, the possibility of selection bias in test ordering within this healthcare system should also be discussed. The discrepancy may reflect differences in who is tested, thresholds for ordering tests, care settings, or clinical pathways rather than true differences in infection rates. Additional stratified analyses, if possible, or a more explicit framing of this observation as hypothesis-generating would strengthen the interpretation.

The handling of atypical outliers is another area that should be treated with explicit statistical caution. The pneumococcal positivity rate in 2007 reportedly falls outside the 1.5×IQR range and is attributed to pre-vaccination epidemiology. While this explanation is plausible, such a baseline-year outlier could influence the definition of “normal” performance and may affect alert thresholds if incorporated without sensitivity analyses. The manuscript would be strengthened by demonstrating robustness of the baseline estimates to inclusion or exclusion of early years, and by clarifying how structural breaks—such as vaccine introduction or changes in clinical practice—are accommodated in the monitoring framework.

The most important limitation remains the absence of linkage between aggregated laboratory positivity and individual clinical outcomes, treatment information, or confirmatory reference testing. Without a gold-standard comparator such as culture, PCR, or clinical adjudication, the framework cannot fully disentangle increased true infections from increases in false positives, nor can it confirm that observed peaks correspond to clinically meaningful disease burden. The manuscript acknowledges this limitation, but the Discussion would benefit from elevating it as a central constraint on causal interpretation and from clarifying that the proposed approach is primarily designed for quality monitoring and early signal detection rather than definitive epidemiological attribution.

Related to interpretability, the binary nature of the DIGIVAL readout creates vulnerability to threshold changes that could mimic epidemiological shifts. If the manufacturer altered the software decision threshold, even slightly, the resulting change in positivity could be misinterpreted as a true disease trend. The authors should describe how instrument versioning, software updates, calibration procedures, and any known changes over the study period were tracked and considered analytically. Moreover, assay biological constraints should be integrated more explicitly into the interpretation. The pneumococcal assay’s reported cross-reactivity with alpha-hemolytic streptococci and the Legionella assay’s limitation to serogroup 1 imply that “positivity rate” is not a pure signal of target-pathogen incidence. These constraints introduce noise and potential bias into longitudinal trends, particularly if the circulation of non-target organisms or non-detected serogroups varies over time.

Several of these concerns could potentially be addressed in a revised version if data access permits. The interpretation of outliers and peaks, such as the Legionella elevations in 2021–2022, would be substantially strengthened by linkage to hospital admission data, clinical diagnoses, imaging, severity markers, and confirmatory microbiology, allowing direct evaluation of whether laboratory signals correspond to clinical case burden. The unexpected sex-associated differences in pneumococcal positivity could also be clarified by examining patient-level timelines, including symptom onset, presentation, admission, timing of sample collection, and disease severity, to test the hypothesis that testing occurs at different stages of illness across groups. Even partial linkage for a subset of years or a representative sample could materially enhance the rigor of these interpretations.

Finally, the Discussion would benefit from a clearer articulation of how the proposed epidemiological QA framework could be improved if quantitative outputs were available from point-of-care diagnostics. Access to continuous signal intensity, near-threshold distributions, or kinetic readouts would allow monitoring beyond binary positivity, enabling earlier detection of drift, better separation of threshold shifts from true prevalence changes, and more sensitive identification of subclinical analytical degradation. Emphasizing this forward-looking implication would strengthen the manuscript’s relevance to the future design of rapid diagnostic platforms and to national strategies for sustaining diagnostic reliability over long time horizons.

.

Reviewer #1: No

Reviewer #2: No

---

## [Author Response · Author response to Decision Letter 1]

6 Feb 2026

PONE-D-25-53718

Trend-Based Quality Assurance of Binary Urinary Antigen Tests

Dear Dr Vall-Ilosera Camps,

Please find enclosed the revised version of our manuscript entitled “Trend-Based Quality Assurance of Binary Urinary Antigen Tests.” A detailed, point-by-point response to the editor’s and reviewers’ comments is provided below.

We are grateful for the reviewers’ constructive comments, which have helped us clarify the scope and interpretation of the study and improve the manuscript. Yours sincerely,

Susanne Sütterlin

5. Review Comments to the Author

Reviewer #1: Binary tests (positive/negative) lack the quantitative output typically used in statistical process control (like Shewhart charts). This study addresses a significant gap by using population-level positivity trends as a proxy for assay stability.

Substantial Dataset: The use of over 32,000 tests (17,356 pneumococcal and 15,280 Legionella) across nearly two decades provides high statistical power to distinguish between technical drift and actual epidemiological shifts.

Context-Aware Interpretation: The researchers successfully correlated outliers (e.g., Legionella peaks in 2021–2022) with known epidemiological events rather than dismiss them as test failures, demonstrating the practical utility of their framework in a real-world setting.

Demographic Insights: The study uncovered significant demographic trends, such as higher pneumococcal positivity in children (20.3%) and female patients (8.6% vs. 6.3% in males), which adds secondary value to the primary QA objective.

We thank Reviewer #1 for the positive and thoughtful assessment of the manuscript, highlighting the relevance of the proposed framework for binary assays, the strength of the long-term dataset, and the practical interpretation of population-level trends.

Methodological Considerations

Statistical Robustness: The application of interquartile range (IQR) thresholds (median ±1.5×IQR) is an appropriate choice for "real-world" data that may not follow a normal distribution.

Theoretical vs. Practical Comparison: By including Bayesian predictive values, the authors illustrate the dramatic impact of prevalence on test performance, providing a useful theoretical benchmark for why lab-based monitoring is necessary.

Limitation - Lack of Clinical Linkage: A noted limitation is the absence of individual clinical outcomes. As the study is based on aggregated laboratory data, it cannot directly validate individual test accuracy against a "gold standard" clinical diagnosis.

We thank the reviewer for the positive assessment of the statistical approach and for highlighting the relevance of using IQR-based thresholds and Bayesian predictive values in a real-world laboratory setting. We agree with the reviewer that the absence of linkage to individual clinical outcomes precludes validation of individual test accuracy against a clinical gold standard. However, this is not the aim of the present study. The proposed framework is intended as a trend-based quality monitoring and signal detection approach at the population level, rather than as a method validation or diagnostic accuracy study. Within this intended use, aggregated laboratory positivity trends provide a pragmatic and independent performance signal, and the lack of individual clinical linkage is therefore an acknowledged and inherent limitation rather than a methodological shortcoming.

Conclusion & Impact

The research is highly relevant for clinical laboratories seeking low-cost, effective ways to monitor qualitative assays. It provides a feasible strategy for long-term population-based monitoring that can detect subtle analytical shifts which internal manufacturer controls might miss. The study is well-structured and scientifically sound, offering a practical framework that can be adapted for other binary diagnostic assays beyond urinary antigens.

We thank the reviewer for the positive assessment of the study’s relevance and practical impact.

Reviewer #2: This manuscript addresses an important and underdeveloped aspect of diagnostic medicine, namely how to ensure long-term quality assurance for rapid, reader-based assays that provide only binary outcomes. By analyzing 17 years of laboratory testing data (2007–2024) comprising more than 30,000 assays, the study proposes a practical framework in which population-level longitudinal positivity trends are monitored to identify potential analytical drift and to distinguish true epidemiological variation from technical or operational failures. The work is potentially valuable for Sweden’s healthcare system because it offers stable long-term reference baselines for urinary antigen testing, with reported positivity levels of approximately 7.4% for Streptococcus pneumoniae and 0.7% for Legionella, which could serve as performance anchors for surveillance and rapid troubleshooting.

A central contribution is the demonstration that long-term positivity rates can remain stable despite large increases in test volumes over time. This stability supports the concept that routine clinical data can be used as an independent performance signal, particularly in settings where conventional quality controls may not detect subtle changes in assay behavior. The manuscript also carries a preventive medicine implication by suggesting that national policy interventions, specifically the 2009 introduction of pneumococcal vaccination, coincided with a measurable stabilization of pneumococcal test positivity compared with earlier years. The temporal patterns further underscore the need for seasonal and demographic clinical awareness: pneumococcal positivity appears lowest in August, whereas Legionella positivity varies unpredictably across years, supporting a rationale for continuous longitudinal surveillance to protect vulnerable groups.

The manuscript is especially relevant because it confronts the “black box” problem inherent to binary readouts. Because these assays do not provide graded signal intensity or proximity to the positivity threshold, standard statistical process control approaches are difficult to apply. The authors’ proposal to monitor population-level positivity trends is therefore an innovative and feasible strategy to detect drift or discontinuities that might otherwise go unnoticed. Related to this, the framework may mitigate potential manufacturer bias that can arise when internal controls are supplied by the same manufacturer that produces the test system. By leveraging patient-derived trends and summary statistics, the approach can function as a real-world, independent performance check that complements rather than replaces conventional controls.

We thank Reviewer #2 for the thoughtful and constructive evaluation, highlighting the relevance of population-level trend monitoring for quality assurance of binary diagnostic assays.

Several points would benefit from clarification or strengthening in a revised version. First, the figures appear to be low resolution and are difficult to read; higher-resolution or vector-quality figures are needed so that trends, outliers, and subgroup comparisons can be evaluated reliably.

We thank the reviewer for pointing this out. All figures have now been re-exported and uploaded as separate high-resolution PNG files (600 dpi at final display size) to ensure readability.

Second, the manuscript should more clearly justify the methodological choice of using “observed test positivity” as a prior input when estimating positive predictive values. Because PPV is extremely sensitive to prevalence assumptions, the rationale for treating observed positivity as a proxy for prevalence should be explained in detail, including how the approach avoids circularity and what interpretation the resulting PPV is intended to support. The large discrepancy reported between incidence-based PPV estimates and test-based PPV estimates is striking and deserves careful discussion. The authors should clarify whether the incidence-based approach is using population-level incidence that is not directly comparable to the clinically enriched tested population, and they should contextualize any manufacturer-reported PPV values by describing the underlying study population, reference standard, and case definition. If manufacturer PPV appears low under certain assumptions, it would be helpful to explain whether this reflects test performance limitations or instead reflects the use of prevalence inputs that do not match the intended-use population.

We agree that positive predictive value is highly sensitive to prevalence assumptions. In this study, population-level incidence was included not to estimate diagnostic accuracy, but to demonstrate how different prevalence inputs influence PPV over time and to assess whether temporal changes in PPV estimates aligned with known epidemiological shifts or outbreak-related events. Observed test positivity was used as a longitudinal monitoring signal within the tested population rather than as a proxy for true prevalence. We have clarified this rationale, addressed the issue of circularity, and expanded the discussion to contextualize the observed discrepancies between incidence-based, test-based, and manufacturer-reported PPV estimates. (lines 297-310)

The demographic findings also merit deeper consideration. The observation of higher pneumococcal positivity in females than males contrasts with some published reports suggesting male predominance among hospitalized pneumonia cohorts. While the authors propose that differences in the timing of testing might contribute, the possibility of selection bias in test ordering within this healthcare system should also be discussed. The discrepancy may reflect differences in who is tested, thresholds for ordering tests, care settings, or clinical pathways rather than true differences in infection rates. Additional stratified analyses, if possible, or a more explicit framing of this observation as hypothesis-generating would strengthen the interpretation.

We agree that the observed sex differences in pneumococcal test positivity require cautious interpretation. We have expanded the Discussion to address potential selection bias related to test ordering practices, care settings, and clinical thresholds, and we frame these findings as hypothesis-generating rather than indicative of true sex-specific differences in infection rates. (lines 332-335)

The handling of atypical outliers is another area that should be treated with explicit statistical caution. The pneumococcal positivity rate in 2007 reportedly falls outside the 1.5×IQR range and is attributed to pre-vaccination epidemiology. While this explanation is plausible, such a baseline-year outlier could influence the definition of “normal” performance and may affect alert thresholds if incorporated without sensitivity analyses. The manuscript would be strengthened by demonstrating robustness of the baseline estimates to inclusion or exclusion of early years, and by clarifying how structural breaks—such as vaccine introduction or changes in clinical practice—are accommodated in the monitoring framework.

We agree that atypical early-year outliers and structural breaks require cautious interpretation. We have revised the Discussion to clarify that elevated pneumococcal positivity in early years, including the pre-vaccination period, reflects true epidemiological conditions rather than analytical instability, and that the proposed framework is intended for dynamic, context-aware interpretation rather than fixed thresholding across structurally different periods. (lines 337 – 354)

The most important limitation remains the absence of linkage between aggregated laboratory positivity and individual clinical outcomes, treatment information, or confirmatory reference testing. Without a gold-standard comparator such as culture, PCR, or clinical adjudication, the framework cannot fully disentangle increased true infections from increases in false positives, nor can it confirm that observed peaks correspond to clinically meaningful disease burden. The manuscript acknowledges this limitation, but the Discussion would benefit from elevating it as a central constraint on causal interpretation and from clarifying that the proposed approach is primarily designed for quality monitoring and early signal detection rather than definitive epidemiological attribution.

We agree that the absence of linkage to individual clinical outcomes is a central limitation. We have revised the Discussion to point it out on causal interpretation and to explicitly clarify that the proposed framework is intended for quality monitoring and early signal detection rather than definitive epidemiological attribution or diagnostic accuracy assessment. (lines 332-335)

Related to interpretability, the binary nature of the DIGIVAL readout creates vulnerability to threshold changes that could mimic epidemiological shifts. If the manufacturer altered the software decision threshold, even slightly, the resulting change in positivity could be misinterpreted as a true disease trend. The authors should describe how instrument versioning, software updates, calibration procedures, and any known changes over the study period were tracked and considered analytically. Moreover, assay biological constraints should be integrated more explicitly into the interpretation. The pneumococcal assay’s reported cross-reactivity with alpha-hemolytic streptococci and the Legionella assay’s limitation to serogroup 1 imply that “positivity rate” is not a pure signal of target-pathogen incidence. These constraints introduce noise and potential bias into longitudinal trends, particularly if the circulation of non-target organisms or non-detected serogroups varies over time.

We agree that binary readouts are sensitive to threshold-related effects and that assay-specific biological constraints must be considered when interpreting longitudinal positivity trends. We have expanded the Discussion to address potential effects of instrument and software changes, as well as known biological limitations of the pneumococcal and Legionella assays, and to clarify that observed positivity rates are interpreted as quality monitoring signals rather than direct measures of pathogen incidence. (lines 271-278)

Several of these concerns could potentially be addressed in a revised version if data access permits. The interpretation of outliers and peaks, such as the Legionella elevations in 2021–2022, would be substantially strengthened by linkage to hospital admission data, clinical diagnoses, imaging, severity markers, and confirmatory microbiology, allowing direct evaluation of whether laboratory signals correspond to clinical case burden. The unexpected sex-associated differences in pneumococcal positivity could also be clarified by examining patient-level timelines, including symptom onset, presentation, admission, timing of sample collection, and disease severity, to test the hypothesis that testing occurs at different stages of illness across groups. Even partial linkage for a subset of years or a representative sample could materially enhance the rigor of these interpretations.

We agree that linkage to individual-level clinical data would strengthen causal interpretation. We have revised the Discussion to elevate the absence of such linkage as a central limitation and to explicitly clarify that the framework is designed for quality monitoring and early signal detection rather than causal or epidemiological attribution. (lines 337-354)

Finally, the Discussion would benefit from a clearer articulation of how the proposed epidemiological QA framework could be improved if quantitative outputs were available from point-of-care diagnostics. Access to continuous signal intensity, near-threshold distributions, or kinetic readouts would allow monitoring beyond binary positivity, enabling earlier detection of drift, better separation of threshold shifts from true prevalence changes, and more sensitive identification of subclinical analytical degradation. Emphasizing this forward-looking implication would strengthen the manuscript’s relevance to the future design of rapid diagnostic platforms and to national st

---

## [Editor Report · Decision Letter 1]

26 Feb 2026

Trend-Based Quality Assurance of Binary Urinary Antigen Tests

PONE-D-25-53718R1

Dear Dr. Sütterlin,

We’re pleased to inform you that your manuscript has been judged scientifically suitable for publication and will be formally accepted for publication once it meets all outstanding technical requirements.

Kind regards,

Rajeev Singh

Academic Editor

PLOS One
---

## [Editor Report · Acceptance letter]

PONE-D-25-53718R1

PLOS One

Dear Dr. Sütterlin,

I'm pleased to inform you that your manuscript has been deemed suitable for publication in PLOS One. Congratulations! Your manuscript is now being handed over to our production team.

Kind regards,

on behalf of

Dr. Rajeev Singh

Academic Editor

PLOS One